# Diversity of Mycorrhizal Fungi in Temperate Orchid Species: Comparison of Culture-Dependent and Culture-Independent Methods

**DOI:** 10.3390/jof10020092

**Published:** 2024-01-23

**Authors:** Sophie Mennicken, Caio César Pires de Paula, Hélène Vogt-Schilb, Jana Jersáková

**Affiliations:** 1Department of Ecosystem Biology, Faculty of Science, University of South Bohemia, Branišovská 1760, 37005 České Budějovice, Czech Republic; mennis01@prf.jcu.cz (S.M.); cpiresdepaula@prf.jcu.cz (C.C.P.d.P.); helene.vogtschilb@gmail.com (H.V.-S.); 2Institute of Hydrobiology, Biology Centre CAS, 37005 České Budějovice, Czech Republic; 3Centre d’Écologie Fonctionnelle et Évolutive (CEFE), Centre National de la Recherche Scientifique (CNRS), Université de Montpellier, École Pratique des Hautes Études (EPHE), Institut de Recherche pour le Développement (IRD), 1919 Route de Mende, 34293 Montpellier, France

**Keywords:** Orchidaceae, mycorrhizal fungi, Tulasnellaceae, Ceratobasidiaceae, Serendipitaceae, metabarcoding, culture-independent and -dependent methods, fungal phylogeny

## Abstract

Many orchid species are endangered due to anthropogenic pressures such as habitat destruction and overharvesting, meanwhile, all orchids rely on orchid mycorrhizal fungi (OMF) for seed germination and seedling growth. Therefore, a better understanding of this intimate association is crucial for orchid conservation. Isolation and identification of OMF remain challenging as many fungi are unculturable. In our study, we tested the efficiency of both culture-dependent and culture-independent methods to describe OMF diversity in multiple temperate orchids and assessed any phylogenetic patterns in cultivability. The culture-dependent method involved the cultivation and identification of single pelotons (intracellular hyphal coils), while the culture-independent method used next-generation sequencing (NGS) to identify root-associated fungal communities. We found that most orchid species were associated with multiple fungi, and the orchid host had a greater impact than locality on the variability in fungal communities. The culture-independent method revealed greater fungal diversity than the culture-dependent one, but despite the lower detection, the isolated fungal strains were the most abundant OMF in adult roots. Additionally, the abundance of NGS reads of cultured OTUs was correlated with the extent of mycorrhizal root colonization in orchid plants. Finally, this limited-scale study tentatively suggests that the cultivability character of OMF may be randomly distributed along the phylogenetic trees of the rhizoctonian families.

## 1. Introduction

The Orchidaceae is one of the largest plant families, comprising approximately 28,000 species [1,2,3]. It is also one of the most threatened globally, with an estimated 57% of orchid taxa considered endangered based on assessments using IUCN Red List criteria (ca. 1000 species) [4,5,6]. Orchid seed germination and subsequent seedling development are entirely reliant on a symbiotic relationship with fungi, wherein the fungi trade water and minerals for plant photosynthates [7]. This dependence underscores the importance of suitable fungal associates for the establishment and maintenance of orchid communities and populations [8]. Thus, a more comprehensive knowledge of their root-associated fungal taxa is needed for restoration and conservation projects [6].

Most green orchids are associated with rhizoctonia fungi [9,10], which belong to a polyphyletic basidiomycetous group that includes the Tulasnellaceae, Ceratobasidiaceae (Cantherellales) and Serendipitaceae families (Sebacinales group B order) [11,12,13]. Members of all three families display various ecological niches, living as soil saprotrophs (i.e., decomposers of organic matter, [14]), orchid mycorrhizal symbionts or plant pathogens, and they are known to be globally distributed [15]. In addition to rhizoctonia, orchid roots also harbour various endophytic fungi [13,16,17] and ectomycorrhizal (ECM) fungi from the Basidiomycota and Ascomycota phyla [18,19], some with known mycorrhizal abilities in forest-dwelling orchids [9,20]. Orchid mycorrhizal fungi (OMF) colonize germinating seeds and root cortical cells of adults, forming highly coiled structures called pelotons [10]. As the nutritional exchange occurs across the mycorrhizal interface or after the degeneration and lysis of pelotons in the orchid root cells, the formation of pelotons is an important proof of the mycorrhizal ability of the fungus [5].

Orchid species can associate with multiple fungal taxa simultaneously [21]. However, the diversity of fungal associates appears to be dependent on the host specificity, which varies across orchid species. The orchid mycorrhizal richness, defined as the number of fungal operational taxonomic units (OTUs) associating with a given plant species over its distribution, varies widely. Associations range from narrow, involving only a few OTUs, to broader associations encompassing many OTUs from different fungal groups [22,23,24]. It has been suggested that mycorrhizal specificity might be influenced by environmental conditions [25,26], leading to the presence of different fungal taxa in a same orchid species across different sites. However, how various abiotic and biotic factors contribute to fungal specificity is still not well understood.

In temperate grassland ecosystems, orchid-OMF associations serve as an ‘ideal model’ for research on the biodiversity and evolution of interactions because rhizoctonia fungi are well cultivable under laboratory conditions [9]. Unlike arbuscular mycorrhizal fungi (Glomeromycota phylum) or ECM fungi, which are obligate symbionts and difficult to cultivate without a host plant [27,28], rhizoctonia fungi are not obligate symbionts. Traditionally, culture-dependent techniques and the characterization of morphological characters in a culture have been employed to identify the fungal taxa associated with orchid species. Furthermore, these methods are used to test the functional role and implications for seed germination [10]. This approach is still used worldwide [29], but its results underestimate the overall diversity of OMF since it is a selective method and is subject to factors such as surface sterilization techniques, culturing media and incubation conditions, mostly employed to avoid bacterial contamination [29]. This has led to the adoption of barcoding techniques such as Sanger sequencing and next-generation sequencing (NGS), which have significantly improved the identification of root-associated fungi in recent decades [30]. The internal transcribed spacers of the rDNA gene (ITS1-5.8S-ITS2) have become the most commonly sequenced region for the identification of fungi due to its high variation [31]. Besides mycorrhizal fungi, culture-independent techniques also identify endophytes with roles in plant nutrition not well studied yet, mostly recognized as fungal pathogens, but with increasing evidence of benefits for plant protection and growth, even in some epiphytic orchids (see [32]). When applied alone, the culture-independent approach suffers from biases [33], such as primer mismatches [34] or primer cross-contaminations [35]. This underscores the potential benefit of combining both methods to improve fungal diversity estimation and still be aware of the real contribution of particular OMF to germination and orchid nutrition. However, only a few studies have combined both approaches [17,36].

In this study, we compared a culture-dependent approach (involving the cultivation and identification of single pelotons) with a culture-independent one (direct molecular identification of root mycobionts) to characterize the diversity and cultivability of OMF in 19 temperate orchid species occurring in four species-rich grassland communities in southern France and the Czech Republic. We aimed to (i) describe OMF diversity in multiple orchid species and sites, (ii) compare the efficiency of both methods in determining the diversity of OMF and (iii) identify fungi detected by the culture-independent method but absent in a culture-dependent approach. We asked following questions: What drives OMF community diversity: orchid host identity or locality? Are the dominant fungi of roots identified by the culture-independent method the same as those obtained through cultivation? Can the number of OTU reads (NGS data) accurately assess the proportion of mycorrhizal root colonization by cultured OTUs? Is the cultivability of a fungus linked to its phylogenetic position?

## 2. Materials and Methods

### 2.1. Orchid Species and Root Collection

In May–June 2018, we collected three to five roots per flowering plant from 309 individuals of 19 orchid species located in species-rich mesic grasslands in the Czech Republic, referred to as CZ1 and CZ2 (49°7′ N, 13°39′ E and 48°53′ N, 17°31′ E), and in southern France, referred to as FR1 and FR2 (43°53′ N, 3°15′ E and 43°58′ N, 3°24′ E) (Table 1). The current study is an extension of the previous sampling focused on the coexistence of orchid communities in species-rich grassland habitats [37]. The four sites had similar soil parameters, though the climatic conditions differed (Mediterranean climate with dry summers vs. continental climate of Central Europe; see details in [37]). For fungus isolation and identification, roots were transported to the lab in moist plastic bags, placed in the fridge and subjected to both culture-dependent and culture-independent methods in the following days.

### 2.2. Root Preparation for Identification of Fungal-Root Associates

Roots were rinsed to remove soil and debris and were surface-sterilized for 30 s in 4.7% sodium hypochlorite, followed by three 30 s rinse steps in sterile distilled water. The extent of mycorrhizal colonization was estimated under a stereomicroscope by sectioning each root into 0.5–1 cm segments. We assigned a percentage of mycorrhizal colonization to cross sections (1 mm wide) of each root segment into five categories corresponding to 0% (category 0), 0–25% (1), 25–50% (2), 50–75% (3) and more than 75% (4). From each colonized root, we selected, on average, 15 pelotons for culture preparation (‘culture-dependent’). Twelve well-colonized root sections (3 mm wide) per plant were stored in cetyl trimethylammonium bromide (CTAB) for further molecular identification within a month via NGS techniques (‘culture-independent’).

### 2.3. Culture-Dependent and -Independent Assessment of OMF

For culture-dependent assessment of OMF, individual pelotons were extracted from the cortical cells of several cross sections of a root. The pelotons were rinsed four times in distilled water, micropipetted onto modified Melin-Norkrans (MMN) medium with Novobiocin and kept at 20 °C in the dark. In the following days, growing hyphal tips were subcultured into new MMN Petri dishes without antibiotics and kept at 20 °C until the colony reached a sufficient size for mycelium sampling. DNA extraction was performed using the NaOH method [38]. The internal transcribed spacer (ITS) of nrDNA was amplified using three pairs of primers: ITS1/ITS4 [39], ITS1OF/ITS4OF [40] or ITS1/ITS4TUL, following the protocol used in Těšitelová et al. (2013) [23]. The amplicons were Sanger sequenced by SeqMe company (Dobříš, Czech Republic). The sequences were grouped into operational taxonomic units (OTUs) at a commonly used 97% similarity threshold over the ITS region.

For the culture-independent assessment of OMF, DNA was initially extracted using a CTAB-based method [41]. Subsequently, amplicons were obtained using two pairs of primers (5.8S-OF/ITS4OF and 5.8S-OF/ITS4TUL, as described in [24]) that cover the entire diversity of rhizoctonian OMFs [42]. MiSeq Illumina sequencing was conducted by the SEQme Company (Dobříš, Czech Republic). Bioinformatic procedures were applied after (i.e., sequence’s merging, quality thresholds, removal of singletons and putative chimeras, alignments) (see [37] for further details). The sequences were first grouped into putative OMF taxa at a similarity of ≥85% if they belonged to the rhizoctonian families (Ceratobasidiaceae, Tulasnellaceae and Serendipitaceae) and thus separated from other ectomycorrhizal and saprotrophic fungi. Subsequently, sequences were grouped into OTUs based on a threshold of 95% similarity.

Fungal sequences obtained by the two methods were deposited in GenBank (SUB14014389 and SUB14019687, Appendix A).

### 2.4. Phylogenetic Trees of Fungal OTUs Obtained from Both Methods

Sequences obtained from NGS, and fungal isolates were utilized to identify their closest counterparts in the National Center for Biotechnology Information (NCBI) GenBank database (www.ncbi.nlm.nih.gov/genbank; accessed on 30 August 2023) with a high similarity threshold (≥97%). The sequences of *Saitozyma pseudoflava* (MK050284.1) and *Trichosporon* sp. (DQ288848.2) were designated as outgroup taxa. Phylogenetic analyses were performed separately for each rhizoctonian family (i.e., Tulasnellaceae, Ceratobasidiaceae and Serendipitaceae). Multiple sequence alignments were generated using MAFFT v7.310 with the L-INS-i strategy [43]. Alignments were then visualized and manually trimmed using BioEdit Sequence Alignment Editor [44] to ensure the best common coverage after alignment. The optimum nucleotide substitution model was determined using the jModelTest2 v.2.1.6 on XSEDE [45] based on the Akaike and Bayesian information criteria (AIC and BIC). Phylogenetic trees were then constructed using Bayesian inference (BI). The BI analysis was performed using MrBayes on XSEDE v.3.2.7a [46] through the CIPRES Science Gateway V.3.3 (phylo.org) [47]. The selected optimal model was the gamma distribution with invariant sites (GTR+I+G) model, and Bayesian posterior probabilities (PP) were obtained from the 50% majority-rule consensus of the retained trees. The tree topologies were visualized in FigTree v1.4.4 [48] and edited using MEGA v11 [49].

### 2.5. Data Analysis

All analyses were performed using R software v4.1.0 [50]. The relative frequencies of OTUs obtained by NGS were expressed as the number of reads of an OTU within a sample divided by the total number of reads found in that sample (see [37]).

To evaluate the fungal diversity and community composition across orchid species, we calculated the Shannon diversity index and dissimilarity index (DI) based on Bray–Curtis (BC) distances using the ‘vegdist’ function in the vegan package. The BC distances were subsequently averaged at two different scales to derive fungal community dissimilarity indices between (1) heterospecific plants within a site (hereafter referred to as “interspecies DI”) or (2) conspecific plants occurring at different sites (referred to as “intersite DI”). To assess whether the fungal community composition is influenced by either the orchid host identity or the locality (i.e., site of sampling), we selected only the orchid species that occurred in at least two sites (see Table 1). This analysis was conducted using distance-based redundancy analysis (db-RDA) in the vegan package [51], followed by a permutation test. We performed ANOVA tests on the BC dissimilarity indices to determine if fungal communities significantly differ between orchid species and within the same orchid species occurring at multiple sites.

We conducted ANOVA tests to examine the correlation between the relative frequencies of the reads obtained through NGS and the fungal cultivability trait (i.e., whether the fungi were cultured or not). Additionally, we employed the Pearson test to assess whether the relative abundance of OTUs based on sequence reads (culture-independent method) was correlated with their proportion calculated from the mycorrhizal colonization (culture-dependent method) of corresponding roots for each cultivated fungal OTU isolated from individual plants. To calculate the proportion of OTUs in colonized roots, we first summed up the percentage of mycorrhizal colonization across all the root sections for a given OTU isolated from corresponding roots. Then, we recalculated the proportion of each OTU in the individual plant by dividing the sum of mycorrhizal colonization of the OTU by the total sum of the colonization across all colonized root sections per plant.

## 3. Results

### 3.1. Culture-Independent Method: Diversity and Community Composition of Rhizoctonia across Orchid Species and Sites

The next-generation sequencing revealed a total of 63 putative orchid mycorrhizal OTUs (1,554,761 sequences) associated with 309 orchid plants. The orchids exhibited a predominant association with rhizoctonian OMF belonging to Tulasnellaceae (26 OTUs), Ceratobasidiaceae (19 OTUs) and Serendipitaceae families (9 OTUs) (Figure 1, Appendix A). Additionally, ectomycorrhizal fungi (9 OTUs) with potential mycorrhizal abilities were found to be associated with some orchid species. These ectomycorrhizal fungi belonged to six fungal families: Sebacinaceae (1 OTU), Thelephoraceae (3 OTUs), Omphalotaceae (1 OTU), Psathyrellaceae (1 OTU), Russulaceae (2 OTUs) and Mycenaceae (1 OTU) (refer to Appendix A).

Among the rhizoctonian OTUs, all orchid species were found to associate with multiple OTUs within a site, except for *Gymnadenia densiflora*, which was associated with only one Tulasnellaceae OTU (Figure 1). Notably, five orchid species, *G. conopsea* (diploid), *Neotinea ustulata*, *Neottia ovata*, *Anacamptis pyramidalis* and *A. morio*, exhibited the highest fungal richness per site, ranging from 9.3 to 11.7 fungal OTUs. Most of the other orchid species were associated with five to eight fungal OTUs (Table 1, Figure 1). In contrast, four orchid species—*G. densiflora*, *Orchis mascula*, *O. purpurea* and *Ophrys apifera*—were associated, on average, with only 1 to 4.5 fungal OTUs per site. In terms of fungal family composition, certain orchid species exclusively associated with Tulasnellaceae OTUs, such as *G. densiflora*, *Oph. apifera* and *Oph. holubyana*, while Ceratobasidiaceae OTUs were predominant in the genus *Platanthera*. Serendipitaceae OTUs were particularly dominant in *N. ovata* (Figure 1). Furthermore, most orchid species formed interactions with one or two predominant OTUs, primarily from the Tulasnellaceae family (Figure 1, Appendix A). Five of those dominant fungal associates reached more than 80% abundance in the roots of *O. mascula* (T2), *O. militaris* and *O. purpurea* (T1), *Op. apifera* (T11), *G. densiflora* (T7) or *P. bifolia* (C1) (Appendix A).

At the plant level, we observed a significant difference in the number of fungal OTUs per individual plant among the 19 orchid species (ANOVA, F _(18, 289)_ = 3.15, *p* = 2.49 × 10^−05^, Appendix A). On average, individual plants were associated with two OTUs (overall mean = 2.01), and the variability was mainly due to *A. morio* (mean = 2.96) and *N. ustulata* (mean = 2.86) species, which were significantly associated with the highest number of OTUs per plant compared to *O. mascula*, *O. purpurea*, *O. militaris* and *Platanthera* spp., which associated with the least OTUs per plant (Appendix A).

A comparison of the OMF community composition in the 11 orchid species occurring at two to three sites, regardless of the region (see Table 1), revealed significant effects of both locality and orchid host identity (orchid species) (PERMANOVA, F = 11.06, *p* = 0.001 and F = 10.07, *p* = 0.001, respectively) (Figure 2, Appendix A). The effects remained significant when we tested each factor individually and when we used partial db-RDA by removing the effects of covariates (Appendix A). Orchid host identity explained 43.9% of the variation in fungal communities, while locality explained 8.2%, and these two variables jointly explained 4.9%. The remaining 43% was unexplained by the model. The first db-RDA axis appears to strongly separate the two species *Orchis militaris* and *O. purpurea*, occurring at the French sites, and another group of five species found at both French and Czech sites: *Dactylorhiza sambucina*, *G. conopsea* (diploid), *N. ovata*, *N. ustulata* and *Platanthera bifolia* (Figure 2). A clear regional separation is evident, with the majority of plants from the Czech sites forming a distinct group, while the plants from the French sites exhibit a more dispersed distribution. The second axis primarily separates two species, *O. mascula* and *A. pyramidalis* (Figure 2).

All orchid species occurring in at least two sites displayed high intersite dissimilarity (mean ± SD = 0.79 ± 0.03) and even higher interspecies dissimilarity indices (0.91 ± 0.06; Figure 3), indicating that the fungal composition of orchid species markedly differed both between conspecific plants from different sites and between heterospecific plants within a same site. The Shannon diversity index showed a significant difference among orchid species (ANOVA, F _(10, 17)_ = 3.44, *p* = 0.012), as did the intersite DI (ANOVA, F _(10, 17)_ = 38.85, *p* = 1.0 × 10^−09^; Appendix A). However, there was no significant difference among species in the interspecies DI (ANOVA, F _(10, 17)_ = 0.71, *p* = 0.71; Appendix A). We also found a significant positive correlation between the Shannon diversity index and the intersite DI (Pearson correlation, *r* (21) = 0.46, *p* = 0.028). This implies that species with lower fungal diversity also exhibited more similar fungal compositions between conspecific plants from different sites (Figure 3). In particular, *A. pyramidalis*, *O. mascula*, *O. militaris* (only plants in French sites) and *O. purpurea* species had different fungal communities compared to the other species. However, their fungal communities were more similar in conspecific plants regardless of the sites (Figure 2 and Figure 3). The species *O. mascula* displayed the lowest Shannon diversity index and hosted a very similar fungal community between the sites (mean intersite DI = 0.18).

When comparing sites from different regions, *N. ustulata* and *N. ovata* emerged as the two orchid species with the highest mean intersite and interspecies DIs, followed by *D. sambucina* (Figure 3). Plants of *A. pyramidalis* shared similar fungal OTUs in both regions (intersite DI = 0.56), whereas plants of *O. militaris* exhibited more similar fungal taxa at the French sites compared to the Czech site (intersite DI = 0.67 and = 0.84, respectively). In contrast, fungal communities in conspecific plants of *A. morio* and *G. conopsea* (diploid) species differed more between sites, in general, than with other heterospecific plants within the same site (Figure 3).

### 3.2. Comparison of OMF Diversity Using Culture-Dependent and -Independent Methods

Using the culture-dependent approach, we were able to isolate and grow pelotons from 172 plants out of 309 sampled plants, resulting in a total of 987 fungal isolates (Appendix A). Among these, 91 plates (9.2%) were identified as contaminants or other endophytic fungi (e.g., *Fusarium* sp.), while the remaining 896 isolates were assigned to 30 rhizoctonia OTUs (Appendix A). The phylogenetic analysis confirmed the taxonomic correspondence between the cultured strains and the OTUs obtained by the culture-independent method (Figure 4). The majority of the obtained isolates belonged to the families Tulasnellaceae (14 OTUs) and Ceratobasidiaceae (17 OTUs), followed by Serendipitaceae (4 OTUs, as shown in Figure 4). The uncultivable OTUs detected by NGS were dispersed across the phylogenetic trees with no clear pattern (Figure 4). Interestingly, three Tulasnellaceae OTUs (e.g., T1, T5, T6) that failed in cultivation across all studied orchids were found in high abundance in the roots (relative frequencies of sequence reads) of *O. militaris*, *O. purpurea*, *O. simia* and *O. anthropophora*, closely clustering in clade A of the Tulasnellaceae tree (Figure 4A, Appendix A). The success of fungus cultivation varied across orchid species, with no OMF cultivated from *O. militaris* and *G. densiflora* species (Figure 5).

When comparing both approaches, the culture-independent method identified a greater overall rhizoctonia richness (number of fungal OTUs per orchid species) than the culture-dependent one. On average, 26.3% of fungal OTUs were detected simultaneously by both techniques, while culture-independent and -dependent techniques alone yielded 73.2% and 0.5% of OTUs, respectively (Figure 5).

The relative frequencies of fungal OTUs obtained through NGS were significantly linked to strain cultivability (ANOVA, F _(1, 270)_ = 64.36, *p* = 3.18 × 10^−14^), as most of the abundant OTUs were successfully cultivated (Figure 6). However, a few fungal OTUs abundant in the roots failed in cultivation and belonged to the family Tulasnellaceae (e.g., T1, T5, T6) (Appendix A, Figure 4A and Figure 6). These uncultivable fungi were mostly found in *O. militaris* and *O. purpurea* species (Figure 5). Some abundant OTUs in a given orchid species could not be cultivated but were isolated from other orchid species. For example, Tulasnellaceae T7, frequent at 100% in *G. densiflora* roots, failed to be cultivated from this species but was isolated from *Neottia ovata* and G. *conopsea* diploid (Figure 5 and Figure 6, Appendix A).

Considering individual plants from which we cultured at least one OTU across all colonized roots (a total of 108 plants), a significantly positive correlation was found between the relative frequencies of the rhizoctonia OTUs (based on the number of sequence reads) and the proportion of mycorrhizal colonization in roots from which the OTU was cultured (Pearson correlation, *r* (225) = 0.74, *p* < 2.2 × 10^−16^, Figure 7), despite data dispersion for certain OTUs. We found 32 isolates reaching 100% abundance in both relative sequence reads and mycorrhizal colonization in roots (Figure 7).

## 4. Discussion

Our study showed that, in general, the orchid species differed greatly in their fungal richness and community composition. The orchid host identity had a greater impact than affiliation with the locality on the variability in root-associated fungal communities. We also provide a comparison of fungal diversity across multiple temperate orchid species using culture-dependent and -independent methods. Though next-generation sequencing provided a better estimate of the overall rhizoctonian diversity in roots, only half of the fungi were cultivable. The cultivability was linked to OTU abundance in NGS reads, which was correlated with the extent of mycorrhizal colonization of roots. Interestingly, some Tulasnellaceae OTUs belonging to clade A failed in cultivation, though they were dominant in orchid hosts (e.g., *Orchis militaris*, *O. purpurea*).

### 4.1. Orchid Mycorrhizal Fungi Diversity among Orchid Species

The culture-independent method identified a relatively high orchid mycorrhizal fungi (OMF) diversity associated with the orchid species. Our data showed that individuals of green orchid species associate simultaneously with multiple fungi, belonging mostly to rhizoctonian OMF, as previously reported from other orchid species-rich communities [52]. We provide substantial evidence that the main rhizoctonian family occurring in all orchid species was Tulasnellaceae [15], while Ceratobasidiaceae and Serendipitaceae OTUs were dominant in *Platanthera* species and *Neottia ovata*, respectively, as previously reported [24,53]. Five orchid species associating with the highest mean number of OTUs per site (9.3 to 11.7), namely *Neotinea ustulata*, *Gymnadenia conopsea* (diploid), *Anacamptis morio*, *A. pyramidalis* and *N. ovata*, are typically considered as generalists (i.e., having a large spectrum of associates over their geographical distribution [23,54,55,56]). On the other hand, some orchid species were associated with rather low fungal richness, such as *G. densiflora*, *Ophrys apifera* and *Orchis mascula*, interacting on average with only one to four Tulasnellaceae OTUs per site, with always one OTU being highly dominant. While *O. mascula* is well known to have one of the highest mycorrhizal specificities across Europe [24,25,57], for the former two species, it is necessary to explore additional populations outside of the Czech Republic to confirm their high specificity [23]. One available study [58], performed in a population of *O. apifera* in Liverpool (England), also detected the dominance of Tulasnellaceae OTUs, and one of them (GenBank accession No. KC243933) was identical with our dominant OTU TUL12 (No. OR990586).

Beside the rhizoctonian OTUs, a few plants of *G. conopsea* diploid and *O. militaris* species formed associations with fungi from Psathyrellaceae, Omphalotaceae and Mycenaceae families, known for their saprotrophic abilities [9,59]. These fungi were also found in the roots of *Dendrobium officinale* and *Cremastra appendiculata* species growing in tropical forests [60,61]. The orchid *G. conopsea* (diploid) was the only species forming an association with two Russulaceae OTUs, which usually form ectomycorrhizal interactions with trees [62]. This association was also observed in the nearly achlorophyllous orchid species *Limodorum abortivum* [63]. Two other well-known ectomycorrhizal fungal families, Sebacinaceae and Thelephoraceae, were found in ten of our studied species (see Appendix A). Their mycorrhizal role has been confirmed in several tropical achlorophyllous orchid species [21] and in European terrestrial genera *Epipactis*, *Gymnadenia* [26,64,65], *Cephalanthera* [66] and *Neottia* [52,53]. Nevertheless, their implication in seed germination in grassland orchid species has not been demonstrated yet.

### 4.2. Effect of Locality and Host Species Identity on Fungal Composition

Our findings indicate that both orchid host identity and locality significantly contribute to the composition of the fungal community in plants. However, the effect of host species was more pronounced compared to that of locality. This suggests that the orchid’s preference for specific fungal taxa might be stronger than the influence of local environmental factors. The largest variation in the fungal community among orchid hosts occurring at multiple sites was explained by the orchid species *O. mascula*, *O. militaris* and *O. purpurea* sampled in France, as well as *A. pyramidalis* sampled in both regions. These orchids exhibited distinct fungal communities when compared to other species. Simultaneously, conspecific plants hosted similar fungal OTUs regardless of their localization. In contrast, a group of species comprising *D. sambucina*, *G. conopsea* (diploid), *A. morio*, *P. bifolia*, *N. ustulata*, *N. ovata* from both regions and *O. militaris* from one Czech site differed greatly in their fungal communities between sites. This result corroborates a previous study that demonstrated high variability in the fungal composition associated with *G. conopsea* species at different sites [24]. Variation in the fungal composition of an orchid species among sites may be attributed to resource competition within a site and/or local environmental conditions. There is a hypothesis suggesting that orchid species form associations with different fungal taxa to reduce competition within species-rich sites [67]. Local environmental conditions, such as soil texture, the amount of nutrients and organic matter, plant composition and temporal variation [5,24], may affect the available pool of OMF fungi at a site, as demonstrated in orchid genera *Epipactis* and *Neottia* [26,68]. At a large geographical scale, variation in fungal community composition associated with the orchid species *Spiranthes spiralis* was evidenced across a latitudinal gradient in its distribution [69].

The significant effect of locality on the orchid fungal community should be approached cautiously, considering that we compared orchid species across a limited number of sites. Additional sampling in more sites would be necessary to unequivocally confirm our results. For instance, previous studies have indicated that at lower altitudes, both species host and site significantly influence the fungal community composition, whereas at higher altitudes, these effects were not as pronounced [70].

### 4.3. Combining Culture-Dependent and -Independent Methods

The culture-dependent approach identified half (55.6%) of the rhizoctonian OTUs detected by the culture-independent method. We identified a larger number of fungal strains compared to a previous study, which found 15% of OTUs by comparing both methods [17]. Despite the lower efficiency of the culture-dependent method due to the limited cultivability of some rhizoctonia, this method provided isolates of the OTUs that were forming abundant pelotons in the roots of adults and were highly abundant in sequence reads. This includes, for instance, strains from Tulasnellaceae (e.g., TUL4, TUL7, TUL8), Ceratobasidiaceae (CER6, CER16, CER19) and Serendipitaceae (SER4) families found in 13 sampled orchid species. Moreover, we obtained taxonomic correspondence between our fungal OTUs (NGS data) and four additional fungal strains (i.e., TUL10, CER2, CER7 and SEB2) that were isolated in similar orchid hosts in our previous unpublished projects, except for CER7 isolated from *Dactylorhiza majalis*.

The cultivability trait was randomly distributed across the phylogenetic trees of the rhizoctonian families, with some exceptions. Three Tulasnellaceae OTUs (T1, T5, T6), highly abundant in roots and sequence reads of *O. militaris*, *O. purpurea*, *O. simia* and *O. anthropophora* (all belonging to the ‘anthropomorphic’ group of the genus *Orchis*, [71]), failed in cultivation and clustered together in clade A of the Tulasnellaceae tree. This clade contains isolates related to *Tulasnella helicospora*, representing an early diverging lineage in some other phylogenies of Tulasnellaceae in basal position of the trees [23,72]. So far, *T. helicospora* was found dominant in the roots of several members of the genus *Orchis* [57,73], and its distribution in soil has been reported in western and central Europe and South America [73,74]. Recently, Calevo et al. (2020) [73] cultivated a *T. helicospora* strain from the rare species *Orchis patens* (GenBank accession No. MT489316, showing 97.9% similarity with our TUL4 strain No. MZ503004) and used it for the successful germination of *O. patens* and *O. provincialis* seeds. We found that the TUL4 strain was highly abundant in the roots of *O. mascula*, as shown in previous studies [24,25]. In the case of *O. militaris*, we did not succeed in culturing any strain despite repeated isolation attempts on different types of cultivation media (MMN, oatmeal, potato-dextrose) enriched in vitamins (such as thiamine and para-amino benzoic acid (PABA)) that have been shown to support the growth of Tulasnellaceae strains [75]. In several cases, we observed initial hyphal growth from a peloton that ended shortly. We found a high similarity of our T1, T5, T6 Tulasnellaceae OTUs with some sequences in the GenBank database: T6, dominant in the roots of *O. militaris*, was 98% similar to Tulasnellaceae OTUs isolated from the same orchid host (EU195344 and GQ907266 [76,77,78]); T5 and T1, dominant in *O. anthropophora*, had a 97% similarity to Tulasnellaceae OTUs isolated from the same species (GQ907250 [77] and GQ907260 [57,77], respectively). According to literature survey, none of the above-mentioned ‘anthropomorphic’ *Orchis* has ever been symbiotically germinated *in vitro*, supporting the limited cultivability of the corresponding Tulasnellaceae strains. Interestingly, two strains from Tulasnellaceae clade A (i.e., TUL1 and TUL4) that were tested in vitro for their nutritional uptake abilities grew more slowly, produced a lower amount of biomass and utilized a narrower spectrum of nutrients compared to isolates from clade B (TUL7 and TUL8) [79]. In addition, we failed to culture the only dominant OTU T7 in *G. densiflora*, but we succeeded in cultivating it from a sister species, *G. conopsea* diploid, from the same site, and we found it to be dominant in the roots of this latter species.

The successful isolation of orchid mycorrhizal fungi into axenic culture and their resynthesis with orchid seeds is an essential step for orchid conservation efforts [2,29]. It has been previously demonstrated that not all endophytic fungi isolated from the roots of adults are able to trigger seed germination. Therefore, it is recommended to isolate fungi from in situ developing protocorms or through seedling-trap experiments [80,81]. In this study, we have shown that the majority of isolated fungi were dominant in both adult roots and the metabarcoding dataset. Interestingly, the same isolates successfully germinated seeds of the orchid species from which they were obtained (our unpublished data).

Previous studies have raised concerns about the reliability of using abundance data derived from next-generation sequencing, attributing potential issues to factors such as PCR primer biases [35]. While some research has indicated that the abundance of pooled OTUs is poorly estimated using high-throughput methods [35,82], our findings revealed a significant positive correlation between the relative frequencies of OTUs (i.e., the number of reads) and the proportions of mycorrhizal colonization in the roots of adult plants. Similarly, Tedersoo et al. (2010) [83] observed that the most dominant ectomycorrhizal species found on root tips often exhibit the highest number of sequences in NGS datasets. More recently, Wang et al. (2023) [84] demonstrated that metabarcoding read counts significantly correlate with actual read abundances measured with droplet digital PCR assays in the samples. Thus, our results do not reject the use of abundance data from NGS analysis to highlight important symbionts that often have ecological relevance [35]. Yet, caution should be made when using NGS-based abundance data, since a non-negligible data dispersion was found between the two methods in this study.

### 4.4. Perspectives for Future Research

While this study has provided valuable insights into orchid–OMF interactions and rhizoctonia cultivability, there are several avenues for further exploration that could enhance our understanding. Limited cultivability of peloton-forming fungi in orchid roots could be overpowered by the exploration of OMF demands on isolation media composition. One main gap in our current knowledge on OMF is in the ecology of enzyme production under various natural or semi-natural conditions, which could be explored in the future with approaches used for arbuscular mycorrhizae [85,86]. Further studies on functional traits of OMF (e.g., growth rate, survival and regeneration) can be evaluated in in vitro experiments [79].

Sampling orchid roots at different periods of the vegetative season could possibly provide pelotons of higher quality for cultivation and better screening of mycorrhizal symbionts with strong seasonality. Orchid roots were shown to be colonized by different fungal taxa in summer and autumn [17], though these patterns seem species-specific [68]. In their recent study, Lespiaucq et al. (2021) [87] reported common temporal turnover of orchid mycobionts, mostly as a partial replacement due to fidelity for a core-group of mycobionts, functional turnover and environmentally driven turnover. Multiseasonal greenhouse experiments or field experiments with re-sampling the same individuals over different vegetative seasons could increase our understanding of a temporal shift in orchid-OMF associations (as also suggested in [87]).

Another not yet well-explored avenue is plant competition for symbionts in species-rich communities which could be potentially targeted by greenhouse pot experiments in which the amount of nutrients and symbiont community can be controlled. A recent meta-analysis based on greenhouse and field experiments dealing with arbuscular mycorrhizae (AMF) has shown that plant competitive ability for AMF depends on plant life history (i.e., perennial, annuals), functional types and abiotic factors [88].

The frequent occurrence of non-rhizoctonia fungi (i.e., ectomycorrhizal Mycenaceae, Thelephoraceae and Russulaceae families or dark septate endophytes) in orchid roots suggest their potential role which is not fully understand yet. Up to now, it has not been proven that they could trigger seed germination processes in tuberous orchids of grasslands, but they were reported to enhance biomass of seedlings and drought resistance in in vitro cultivation experiments [89]. Exploring the role of fungal endophytes in mediating ecological adaptation of orchids to stressful environments might be crucial also for orchid conservation in a changing climate.

## Figures and Tables

**Figure 1 jof-10-00092-f001:**
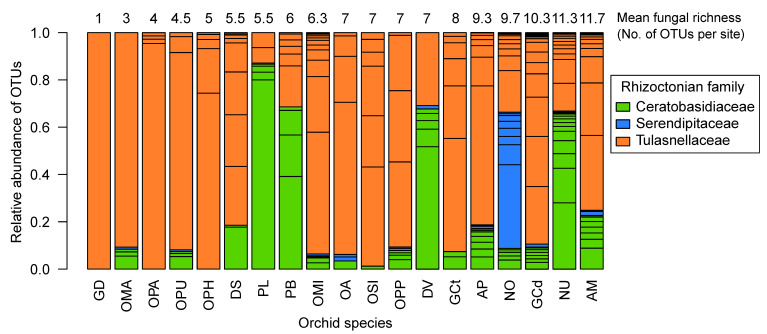
Rhizoctonian OTU diversity and relative frequencies of reads among the 19 studied orchid species. The numbers above the barplots indicate the mean fungal richness per site and orchid species. Rhizoctonian families are colour-coded: Ceratobasidiaceae (in green), Serendipitaceae (in blue) and Tulasnellaceae (in orange). Orchid species: *Anacamptis morio* (AM), *A. pyramidalis* (AP), *Dactylorhiza sambucina* (DS), *D. viridis* (DV), *Gymnadenia conopsea* diploid (GCd), *G. conopsea* tetraploid (GCt), *G. densiflora* (GD), *Neottia ovata* (NO), *Neotinea ustulata* (NU), *Ophrys apifera* (OPA), *Op. holubyana* (OPH), *Op. sphegodes* subsp. *passionis* (OPP), *Orchis anthropophora* (OA), *O. mascula* (OMA), *O. militaris* (OMI), *O. purpurea* (OPU), *O. simia* (OSI), *Platanthera bifolia* (PB) and *Platanthera* spp. (PL).

**Figure 2 jof-10-00092-f002:**
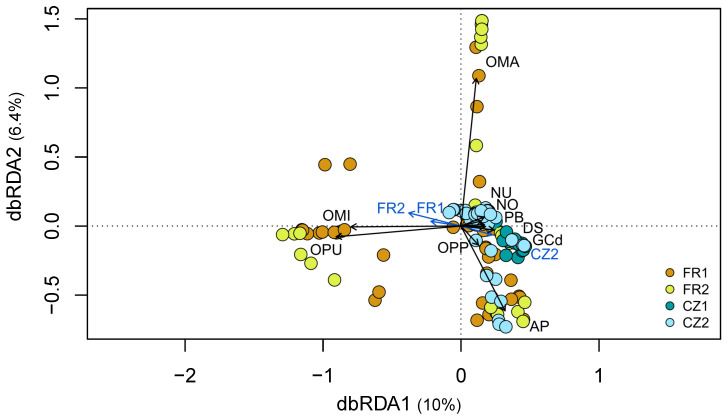
Distance-based redundance analysis comparing the fungal communities in plants of 11 orchid species that occurred at least in two sites. The circles in the diagram represent plants from different orchid species, while the arrows depict explanatory variables: locality (in blue) and orchid host identity (in black). The sites are represented by colours: French sites FR1 (brown) and FR2 (light green), Czech sites CZ1 (dark blue) and CZ2 (light blue). Orchid species: *Anacamptis morio* (AM); *A. pyramidalis* (AP); *Dactylorhiza sambucina* (DS); *Gymnadenia conopsea* diploid (GCd); *Ophrys sphegodes subsp. passionis* (OPP); *Orchis mascula* (OMA); *O. militaris* (OMI); *O. purpurea* (OPU); *Neottia ovata* (NO); *Neotinea ustulata* (NU); and *Platanthera bifolia* (PB).

**Figure 3 jof-10-00092-f003:**
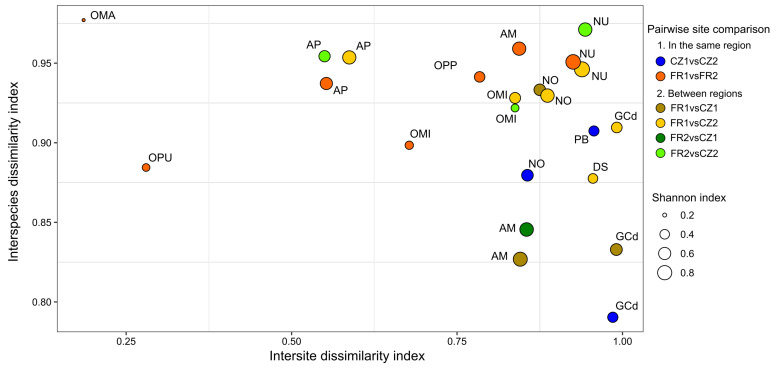
Diversity and dissimilarity indices in OMF community composition of 11 orchid species occurring at multiple sites. The *X*-axis represents intersite dissimilarity indices within the same orchid species across different occurrence sites. The *Y*-axis illustrates interspecies dissimilarity indices between various orchid species coexisting within the same site. Each circle symbolizes a pairwise comparison of fungal dissimilarity indices in the same orchid species at two sites, denoted by the colour of circles, either within or between regions. The size of each circle is proportional to the mean Shannon diversity index of the orchid species at both sites. Orchid species include *Anacamptis morio* (AM); *A. pyramidalis* (AP); *Dactylorhiza sambucina* (DS); *Gymnadenia conopsea* diploid (GCd); *Neottia ovata* (NO); *Neotinea ustulata* (NU); *Ophrys sphegodes* subsp. *passionis* (OPP); *Orchis mascula* (OMA); *O. militaris* (OMI); *O. purpurea* (OPU); and *Platanthera bifolia* (PB).

**Figure 4 jof-10-00092-f004:**
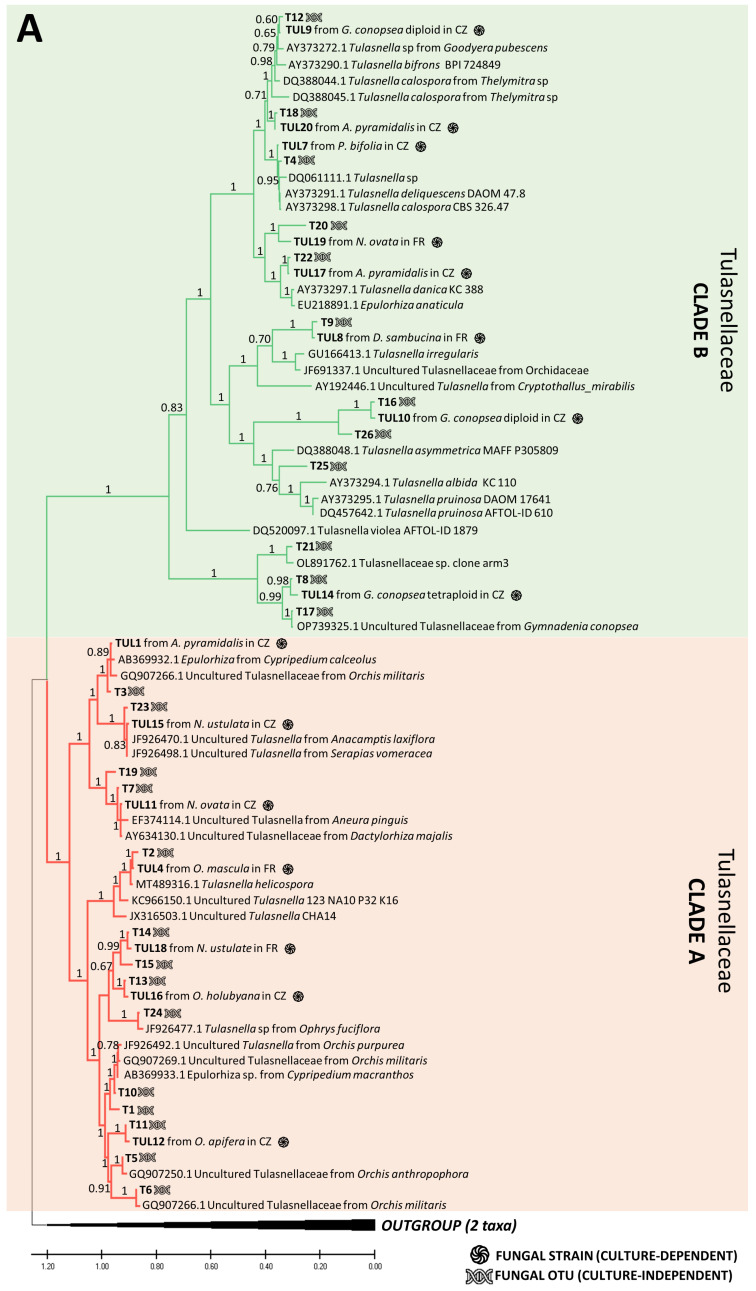
Phylogenetic trees combining the fungal OTUs obtained by culture-dependent and -independent methods. Rhizoctonian families include Tulasnellaceae (two lineages, clades A (red section) and B (green section)) (**A**), Ceratobasidiaceae (**B**) and Sebacinales order (two families, Serendipitaceae (orange section) and Sebacinaceae (blue section)) (**C**). In the illustration, the symbol of DNA helix refers to the fungal OTUs obtained with NGS method, while the black circle represents the fungal OTUs cultivated from pelotons. Additional information on fungal strains includes details about the orchid host and the site of occurrence, indicated as France (FR) or the Czech Republic (CZ) within the tree. Orchid species include *Anacamptis morio*; *A. pyramidalis*; *Dactylorhiza sambucina*; *D. viridis*; *D. majalis*; *Gymnadenia conopsea* diploid; *G. conopsea* tetraploid; *Neottia ovata*; *Neotinea ustulata*; *Ophrys apifera*; *Op. holubyana*; *Op. sphegodes* subsp. *passionis*; *Orchis mascula*; and *Platanthera bifolia*.

**Figure 5 jof-10-00092-f005:**
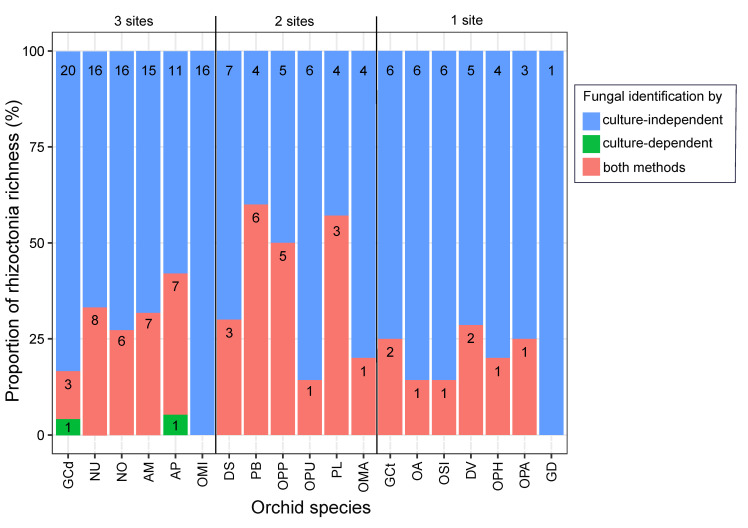
Rhizoctonia richness of orchid species obtained by culture-dependent and -independent methods. Barplots are ranked according to the number of sites of occurrence (1 to 3 sites). The colours indicate the proportion of OTUs detected using different methods: solely through culture-independent methods (in blue), exclusively through culture-dependent methods (in green) and through both methods (in pink). The numbers within the bars correspond to the count of rhizoctonia OTUs. Orchid species include *Anacamptis morio* (AM); *A. pyramidalis* (AP); *Dactylorhiza sambucina* (DS); *D. viridis* (DV); *Gymnadenia conopsea* diploid (GCd); *G. conopsea* tetraploid (GCt); *G. densiflora* (GD); *Neottia ovata* (NO); *Neotinea ustulata* (NU); *Ophrys apifera* (OPA); *Op. holubyana* (OPH); *Op. sphegodes* subsp. *passionis* (OPP); *Orchis anthropophora* (OA); *O. mascula* (OMA); *O. militaris* (OMI); *O. purpurea* (OPU); *O. simia* (OSI); and *Platanthera bifolia* (PB); *Platanthera* spp. (PL).

**Figure 6 jof-10-00092-f006:**
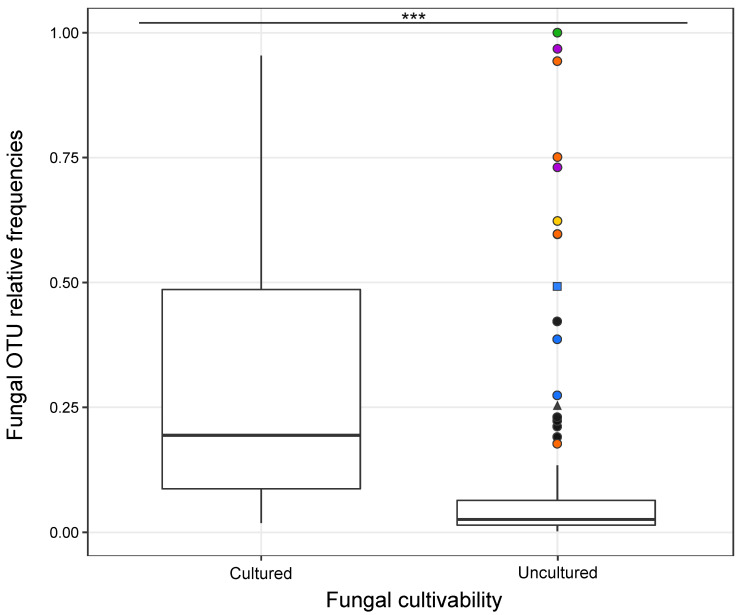
Mean relative frequencies of sequence reads of rhizoctonian fungal OTUs categorized by their cultivability under laboratory conditions (‘cultured’ or ‘uncultured’). The box plots represent median, upper and lower quartiles with the whiskers showing minimum and maximum values, outliers as circles. Different colours and symbols denote abundant OTUs that failed in cultivation from some orchid hosts: *Gymnadenia densiflora* (in green); *Orchis purpurea* (in purple); *O. militaris* (in orange); *O. anthropophora* (in yellow); *Dactylorhiza sambucina* (in blue); and other species (in black). The symbols indicate the fungal family: circle, square and triangle for Tulasnellaceae, Ceratobasidiaceae and Serendipitaceae OTUs, respectively. The symbol (***) indicates a significant difference in the relative frequencies of fungal OTUs between the cultivability status (‘cultured’ versus ‘uncultured’) of the strains, with *p* values ≤ 0.001 based on ANOVA test.

**Figure 7 jof-10-00092-f007:**
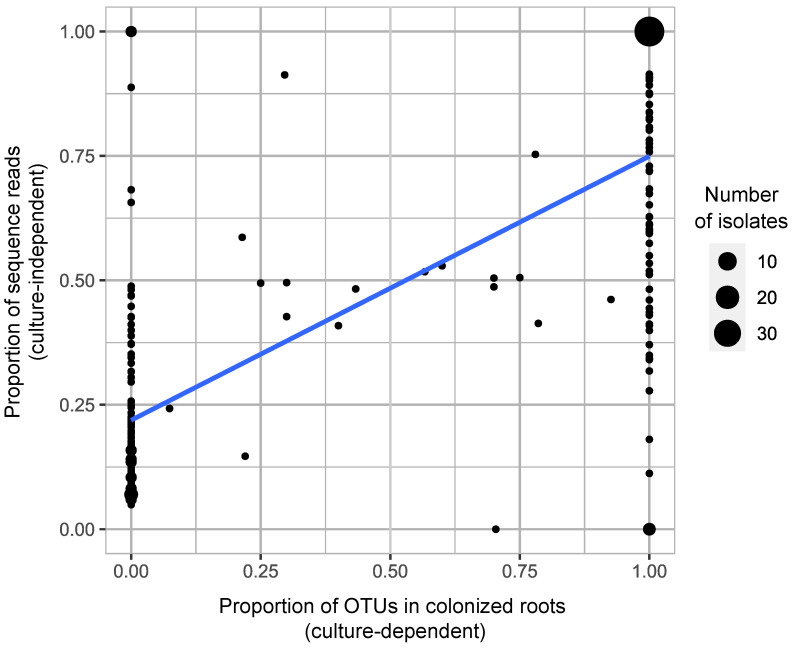
Correlation between the relative frequencies of sequence reads of the cultivable OTUs and their proportion in corresponding mycorrhizal roots. Each dot represents a cultivable OTU’s isolate detected through both culture-dependent and -independent methods across a total of 108 plants. The circles are proportional to the number of isolates. The blue line corresponds to the linear regression model.

**Table 1 jof-10-00092-t001:** Orchid species and genera sampled in regions of the Czech Republic (CZ1 and CZ2) and southern France (FR1 and FR2) with their associate OMF diversity. The symbol (*) indicates the seven orchid species that occurred at two or three sites. The numbers of plants sampled and the total number of rhizoctonian OTU, representing fungal richness, were obtained through both culture-dependent and culture-independent methods.

Orchid Genera	Orchid Species (Code)	No. of Sampled Plants	No. of Fungal OTUs	Sites
*Anacamptis*	*A. morio* (AM) *	27	22	CZ1, FR1, FR2
	*A. pyramidalis* (AP) *	33	18	CZ2, FR1, FR2
*Dactylorhiza*	*D. sambucina* (DS) *	14	10	CZ1, FR1
	*D. viridis* (DV)	5	7	FR1
*Gymnadenia*	*G. conopsea*, diploid (GCd) *	28	23	CZ1, CZ2, FR1
	*G. conopsea*, tetraploid (GCt)	9	8	CZ2
	*G. densiflora* (GD)	4	1	CZ2
*Neottia*	*N. ovata* (NO) *	20	22	CZ1, CZ2, FR1
*Neotinea*	*N. ustulata* (NU) *	22	24	CZ2, FR1, FR2
*Ophrys*	*Op. apifera* (OPA)	6	4	CZ2
	*Op. holubyana* (OPH)	9	5	CZ2
	*Op. sphegodes* subsp. *passionis* (OPP)	17	10	FR1, FR2
*Orchis*	*O. anthropophora* (OA)	9	7	FR2
	*O. mascula* (OMA)	16	5	FR1, FR2
	*O. militaris* (OMI) *	33	16	CZ2, FR1, FR2
	*O. purpurea* (OPU)	18	7	FR1, FR2
	*O. simia* (OSI)	8	7	FR2
*Platanthera*	*P. bifolia* (PB)	15	10	CZ1, CZ2
	*P.* spp. (PL) ^†^	16	7	FR1, FR2

^†^ The French sites contained both orchid species, *Platanthera bifolia* and *P. chlorantha*, with no possibility to correctly distinguish the identity of sterile plants.

## Data Availability

The dataset used in this study is available in the Zenodo Repository DOI https://doi.org/10.5281/zenodo.10507571.

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
