# Peer review of "Diversity of Mycorrhizal Fungi in Temperate Orchid Species: Comparison of Culture-Dependent and Culture-Independent Methods"

_jof, 2024, doi:10.3390/jof10020092_

Round 1

Reviewer 1 Report

Comments and Suggestions for Authors

The goal of the study was to use two methods (culturally dependent and independent) to assess the diversity (and phylogenetic relationships) of orchid mycorrhizal fungi.

The main findings were that most host orchids associated with more than one fungus and that the diversity of fungi was associated with the host plants more than the habitat where the host occurred.

They found that the molecular approach to identifying OMF as more successful than the isolation methods.

The concluded that there was no phylogenetic relationship related to whether or not the OMF could be cultured.

Overall, the manuscript is clearly written, and the graphics are all appropriate for inclusion in the publication.  A few suggestions for additions and issues for the authors to consider are provide below.

The Introduction is very well written and ends with set of questions that are based on a clear statement of the issues related to the identification of orchid OMF and the lack of knowledge on the relationship between the distribution of OMF and the distribution of orchid species at a large geographic scale.

The Methods are clearly presented.

Table 1 shows that some species occurred in both countries and others did not.  It would be helpful if the authors provide more of an explanation behind the approach that was used to sample the orchids. Was it simply based on a goal of sampling each genus in the two countries or some other underlying ideas?   As an example, in lines 176-188 we read about the approach used to conduct the phylogenetic analysis.  Given the importance of this element of the research, why did to authors not restrict their sampling to species that occurred in two or more sites?

Line 199.  Delete ‘studied’

The results clearly show that species and locality are important determinants of OMF but 43% of the variation in the model was not explained – suggesting the importance of other (yet to be determine factors).

An interesting finding was that when OMF were compared for species that occurred at two or more sites, that they differed – further indication that factors that determine what OMF associate with a given orchid species at a site need further evaluation. 

The finding that using a molecular approach to identifying OMF yields a higher diversity than the culture technique is not unexpected, but this is one of the largest studies to document this situation and is thus quite useful and informative.  One underlying issue, however, is that when a root is sampled and the condition of the OMF in the root is also important.  The fact that some pelotons would not be cultured may have simply been due to the physiological condition (i.e., where the elotons still viable?).  Thus, a bit of caution needed in interpreting the results.  In addition, it has been shown that there is a seasonality pattern to the presence of OMF in orchid roots.  The authors sampled once so they likely have a one-time view of the OMF-orchid situation.

Line 379, 391.  Delete ‘studied’

Lines 435-445 give some insight into possible reasons for the observed patterns.  It would be interesting if the authors could provide some guidance on what approaches might be used to gain a better understanding for the patterns and hypothesis (the competition hypothesis).  Also, this is a place in the discussion where the authors might discuss some potential limitations on their research, such as they do note the limited geographic sampling that was done in this project – event though it was quite broad based on most of the previous work that has been done. Another issue not covered is the fact that roots were sampled once and it is know that there are seasonal patterns to OMF (i.e., at times there are no OMF, or viable OMF, in orchid roots).  In addition, I wonder about the relationship between OMF diversity of genera/species and their geographic extent.  Are widespread species more likely to associate with more OMF?  Is there a phylogenetic pattern (e.g., are genera in one part of the orchid family more likely to have a greater diversity of OMF?).

It would be nice to see a section added at the end that provide guidance of what needs to be done to better understand the patterns found in this research.  The Discussion is good in that it provides context for understanding the Results, but it does not do much to further research.  Overall, this is a solid manuscript.  What is demonstrates is no unexpected, but it adds significantly to our understanding or OMF-orchid relationships.

Reviewer 2 Report

Comments and Suggestions for Authors

Included as attachment
